# Sex Differences in Placental Protein Expression and Efficiency in a Rat Model of Fetal Programming Induced by Maternal Undernutrition

**DOI:** 10.3390/ijms22010237

**Published:** 2020-12-28

**Authors:** Sophida Phuthong, Cynthia Guadalupe Reyes-Hernández, Pilar Rodríguez-Rodríguez, David Ramiro-Cortijo, Marta Gil-Ortega, Raquel González-Blázquez, M. Carmen González, Angel Luis López de Pablo, Silvia M. Arribas

**Affiliations:** 1Department of Physiology, Faculty of Medicine, Universidad Autónoma de Madrid, C/Arzobispo Morcillo 2, 28029 Madrid, Spain; sophiph@kku.ac.th (S.P.); cynthia.grh3@yahoo.com (C.G.R.-H.); pilar.rodriguezr@uam.es (P.R.-R.); dramiro@bidmc.harvard.edu (D.R.-C.); m.c.gonzalez@uam.es (M.C.G.); angel.lopezdepablo@uam.es (A.L.L.d.P.); 2Department of Physiology, Faculty of Medicine, Khon Kaen University, Khon Kaen 40002, Thailand; 3PhD Programme in Pharmacology and Physiology, Doctoral School, Universidad Autónoma de Madrid, 28049 Madrid, Spain; 4Department of Medicine, Beth Israel Deaconess Medical Center, Harvard Medical School, 330 Brookline Avenue, Boston, MA 02215, USA; 5Department of Pharmaceutical and Health Sciences, Faculty of Pharmacy, Universidad San Pablo-CEU. C/Julián Romea, 23, Alcorcón, 28003 28925 Madrid, Spain; mgortega@ceu.es (M.G.-O.); raq.gonzalez.ce@ceindo.ceu.es (R.G.-B.)

**Keywords:** corticosterone, oxidative stress, sex, placenta, maternal undernutrition, VEGF

## Abstract

Fetal undernutrition programs cardiometabolic diseases, with higher susceptibility in males. The mechanisms implicated are not fully understood and may be related to sex differences in placental adaptation. To evaluate this hypothesis, we investigated placental oxidative balance, vascularization, glucocorticoid barrier, and fetal growth in rats exposed to 50% global nutrient restriction from gestation day 11 (MUN, *n* = 8) and controls (*n* = 8). At gestation day 20 (G20), we analyzed maternal, placental, and fetal weights; oxidative damage, antioxidants, corticosterone, and PlGF (placental growth factor, spectrophotometry); and VEGF (vascular endothelial growth factor), 11β-HSD2, p22^phox^, XO, SOD1, SOD2, SOD3, catalase, and UCP2 expression (Western blot). Compared with controls, MUN dams exhibited lower weight and plasma proteins and higher corticosterone and catalase without oxidative damage. Control male fetuses were larger than female fetuses. MUN males had higher plasma corticosterone and were smaller than control males, but had similar weight than MUN females. MUN male placenta showed higher XO and lower 11β-HSD2, VEGF, SOD2, catalase, UCP2, and feto-placental ratio than controls. MUN females had similar feto-placental ratio and plasma corticosterone than controls. Female placenta expressed lower XO, 11β-HSD2, and SOD3; similar VEGF, SOD1, SOD2, and UCP2; and higher catalase than controls, being 11β-HSD2 and VEGF higher compared to MUN males. Male placenta has worse adaptation to undernutrition with lower efficiency, associated with oxidative disbalance and reduced vascularization and glucocorticoid barrier. Glucocorticoids and low nutrients may both contribute to programming in MUN males.

## 1. Introduction

Exposure to stress factors during the intrauterine period increases the risk of developing cardiometabolic diseases later in life. In animal models, fetal programming exhibits a sexually dimorphic pattern, with higher susceptibility in males [1]. In a rat model of hypertension programming induced by global maternal undernutrition during gestation (MUN), sex-dependent alterations were already found at weaning, when MUN male offspring exhibits cardiac hypertrophy [2] and a pro-oxidative plasma pattern [3]. However, in female offspring, cardiac alterations were not detected, and females had a higher level of plasma antioxidants.

In humans, there is also evidence that adaptations to the intrauterine and perinatal environment differ in relation to fetal sex. Systematic reviews have shown that women with a male fetus have a higher risk of gestational diabetes [2] and preeclampsia [3]. We have evidence that women carrying a male infant exhibit differences in plasma cytokine pattern, which would endow them with lower capacity to counteract an inflammatory environment [4]. A worse neonatal outcome is also found in males with low birth weight, both in small-for-gestational-age [5] and in premature infants [6,7].

The above-mentioned data in humans and animal models suggest that sex differences in response to fetal stress factors are established early in life. The placenta has the same sex as the fetus and is the interface between the mother and the fetus, being a key determinant in fetal programming. A different placental adaptation to environmental insults, leading to modification in the expression of relevant proteins, has been proposed to contribute to observed differences between males and females [8].

The mechanisms mediating placental sex-specific adaptations to the suboptimal intrauterine environment are still not fully understood. Among them, a deficient vascular development and oxidative stress are plausible, since they are common alterations in pathologies related to placental dysfunction. The two may also be connected, since inadequate placental vascular growth may lead to ischemia–reperfusion and consequent release of excess reactive oxygen species (ROS). Since the fetus has low antioxidant capacity, this may negatively affect fetal growth and development [9].

Several studies have suggested that excess maternal glucocorticoids (GCs) interfere with placental functioning. The mechanisms proposed include increased ROS production, reduced mitochondrial function, decreased expression of antioxidant enzymes, and impaired vascularization [10,11,12,13]. Additionally, high maternal GCs could lead to increased access to the fetus, interfering with growth. Under physiological conditions, this is limited by the placental barrier 11β-hydroxysteroid dehydrogenase type 2 (11β-HSD2), which inactivates active forms of GCs [14]. However, reduced activity and expression has been observed in maternal stress conditions, including maternal malnutrition [15,16,17]. Besides, there are sexually dimorphic differences in placental 11β-HSD2 in response to maternal GCs, which suggest a better adaptation in females [17,18]. Thus, GCs might participate in placental adaptations to adverse intrauterine exposure in a sex-specific manner.

We hypothesize that higher susceptibility to suboptimal nutrition observed in MUN male offspring may be related to worse adaptation of male placenta. To evaluate this hypothesis and analyze the possible mechanisms implicated, we assessed, in a MUN model, at the end of gestation (gestation day 20; G20): (1) nutrient, corticosterone, and oxidative status in maternal plasma; (2) placental growth, efficiency, and protein expression of key enzymatic systems related to oxidative status, vascular development, and GC barrier; and (3) fetal growth parameters and corticosterone level.

## 2. Results

### 2.1. Maternal Weight and Metabolic Parameters

At G20, maternal total weight and net weight (without conceptus) were significantly lower in MUN compared with those in the control group (Figure 1a,b; *p*-value < 0.0001; *p*-value = 0.048). MUN maternal plasma also showed lower protein content compared with control, without significant difference in glycemia (Figure 1c,d; *p*-value = 0.016; *p* = 0.504). Maternal plasma corticosterone was significantly higher in MUN compared with that in control dams (Figure 1e; *p*-value = 0.035).

### 2.2. Maternal Plasma Oxidative Status

Maternal plasma oxidative stress markers are shown in Figure 2. We did not detect statistical differences in either lipid peroxidation (*p-*value = 0.512) or protein oxidation (*p-*value = 0.203) between MUN and control dams.

Antioxidants in maternal plasma are presented in Figure 3. In comparison with control, MUN dams had significantly higher catalase activity (*p*-value = 0.026), without significant differences in SOD activity (*p*-value = 0.378) and GSH (*p*-value = 0.164) and thiol levels (*p*-value = 0.196).

### 2.3. Fetal and Placental Growth Parameters

Figure 4a,b shows fetal anthropometric parameters. Compared with sex-matched controls, fetal weight was significantly lower in MUN fetuses of both sexes (males, *p*-value < 0.0001; females, *p*-value < 0.0001). Similarly, fetal length was smaller in the MUN group (males, *p*-value < 0.0001; females, *p*-value < 0.0001).

Regarding sex differences, control male fetuses showed significantly larger body weight and length compared with their female counterparts (weight, *p*-value = 0.033; length, *p*-value = 0.0006). However, we did not detect statistical differences in body weight and length between MUN males and females (weight, *p*-value > 0.999; length, *p*-value = 0.365).

Placental weight was not different between MUN and control males (*p*-value > 0.999). However, placenta was significantly smaller in MUN females compared with that in sex-matched controls (*p*-value = 0.022). No differences between sexes were observed in placental weight, either in control (*p*-value > 0.999) or in MUN placentas (*p*-value = 0.488; Figure 4c). MUN males exhibited significantly lower feto-placental ratio compared with control males (*p*-value < 0.0001). However, no significant differences were observed between MUN and control females (*p*-value = 0.380). Regarding sex differences, feto-placental ratio was significantly larger in control males than in control females (*p*-value = 0.003), and no differences were observed between sexes in the MUN group (*p*-value > 0.999; Figure 4d).

In MUN male fetuses, plasma corticosterone was also significantly higher compared with that in sex-matched controls (*p*-value = 0.047), while no differences were observed in MUN females compared with their sex-matched control counterparts (*p*-value > 0.999; Figure 4e).

The expression of 11β-HSD2 was significantly reduced in both male and female placentas, compared with that in their sex-matched controls (males, *p*-value < 0.0001; females, *p*-value < 0.0001). MUN female placenta showed a significantly higher expression level compared with MUN male placenta (*p*-value = 0.041; Figure 4f).

### 2.4. Placental Growth Factor (PlGF) and Vascular Endothelial Growth Factor (VEGF)

To determine whether placental vascularization could be altered by maternal suboptimal nutrition, we examined the PlGF level and VEGF protein expression. PlGF was not significantly different between MUN and control groups, either in male or in female placentas (males, *p*-value = 0.306; females, *p*-value = 0.991; Figure 5a). In MUN male placenta, VEGF expression was significantly lower compared with that in control male placenta (*p*-value = 0.002). However, no significant difference was observed in MUN females compared with their sex-matched counterparts (*p*-value > 0.999; Figure 5b). Regarding sex differences, placentas from MUN females exhibited significantly higher VEGF expression compared with those from MUN males (*p*-value = 0.002).

### 2.5. Placental Oxidative Status

#### 2.5.1. Lipid Peroxidation

MDA + HNE levels tended to be higher in MUN placenta, both in males and in females, without statistical differences (males, *p*-value > 0.999; females *p*-value = 0.321; Figure 6).

#### 2.5.2. Expression of ROS-Producing Enzymes

No significant differences in NADPH oxidase subunit p22^phox^ expression were observed between MUN and control groups in either male or female placentas (males, *p*-value > 0.999; females, *p*-value > 0.999; Figure 7a). Xanthine oxidase (XO) expression was significantly increased in MUN males compared with that in male controls (*p*-value = 0.031). However, XO was significantly decreased in MUN female placentas compared with that in sex-matched controls (*p*-value = 0.001). Regarding sex differences, MUN females exhibited significantly lower XO expression compared with males (*p*-value < 0.0001; Figure 7b).

#### 2.5.3. Expression of ROS-Degrading Enzymes

MUN males tended to have lower expression of placental Cu/Zn-SOD (SOD1) without statistical difference (*p*-value = 0.321; Figure 8a), Mn-SOD (SOD2) expression was significantly lower in MUN males (*p*-value = 0.021; Figure 8b), and Ec-SOD (SOD3) was not different (*p*-value > 0.999; Figure 8c) compared with male controls. Expression levels of the mitochondrial antioxidant uncouple protein 2 (UCP2) (*p*-value = 0.045) and catalase (*p*-value = 0.039) were significantly lower in MUN males compared with those in controls (Figure 8d,e).

MUN females showed no differences in SOD1 (*p*-value= > 0.999; Figure 8a), a tendency towards lower SOD2 without statistical difference (*p*-value = 0.188; Figure 8b), and a significantly lower expression of SOD3 (*p*-value = 0.013; Figure 8c) compared with female controls. UCP2 tended to be lower, with no significant differences (*p*-value = 0.233; Figure 8d), and catalase expression was higher in MUN females compared with that in sex-matched controls (*p*-value = 0.019; Figure 8e).

## 3. Discussion

In the present study, we hypothesized that plasticity of the placenta in response to maternal undernutrition exhibits a sexual dimorphic pattern and can modulate fetal programming. This would explain the observed higher susceptibility of males to develop cardiovascular alterations, even in early postnatal life. Our data evidence worse adaptation of male placenta to suboptimal maternal nutrition, with reduced efficiency and downregulation of VEGF, several antioxidant enzymatic systems, and the GC barrier. These data support that male fetus may have inadequate vascularization, further compromising nutrient and oxygen delivery, and may be exposed to higher levels of ROS and maternal GCs. Female placenta responds to reduced maternal nutrition with moderation in placental growth and better angiogenic response, oxidative balance, and GC barrier development, which may reduce their risk of programming. A summary of the main findings and proposed mechanisms implicated is shown in Figure 9.

The placenta, as the interface between the mother and the fetus, is directly influenced by the maternal environment. Therefore, we first analyzed maternal metabolic and oxidative stress alterations, which may influence placental functioning. As expected, by the end of gestation, MUN dams exhibited lower body weight, including weight without conceptus (fetus + placenta). Additionally, they showed lower plasma proteins without alteration in glycemic level. This suggests a maternal metabolic adaptation induced by inadequate nutrition to maintain glucose availability to the growing fetus [19]. The observed low plasma protein levels, together with corticosterone elevation observed in MUN dams, suggest that this hormone was released as a stress response to undernutrition, increasing maternal catabolism (Figure 9), as previously described [20].

In addition to the catabolic effects, increased maternal corticosterone during pregnancy could influence oxidative status. Corticosterone levels appear to be positively associated with ROS production, which may lead to maternal and placental oxidative stress, as previously described in humans [11] and mice [10] exposed to GCs or to stress-induced corticosterone elevation during gestation [21]. During normal pregnancy, there is elevation in ROS production due to increased oxygen demand for fetal growth. Under physiological conditions, this is counteracted by a concomitant rise in antioxidants. However, if this rise does not take place, excess ROS leads to oxidative damage, as observed in mothers with pregnancies complicated with obesity or diabetes [22,23,24]. Our study did not reveal changes in maternal plasma oxidative stress markers, and there was an elevation in catalase enzyme activity. We suggest that could be an adaptation in response to maternal ROS elevation due to both catabolism and high GCs. Catalase was the only enzymatic system elevated in MUN dams. A recent meta-analysis evidences similar findings in women with preeclampsia, showing an increase in catalase serum activity but unchanged or decreased SOD [25]. The differences between catalase and SOD may be related to their response to a high H_2_O_2_ environment. In erythrocytes, catalase has been shown to play a major role in protecting from oxidative damage under high H_2_O_2_ conditions [26]. On the other hand, SOD has been shown to be inactivated by high H_2_O_2_ [27].

As expected, maternal undernutrition induced fetal growth restriction (FGR), evidenced by the lower weight and length in both MUN males and females. Low birth weight is a common finding in different models of exposure to stress factors, including undernutrition, hypoxia, and exposure to excess GCs and toxic substances [1,12]. In control offspring, sexual dimorphism in fetal growth was found, with smaller weight and length in females by G20, as previously described in humans and experimental animals [28,29]. In contrast, no differences between males and females were found in MUN fetuses. These data suggest that the larger growth in males under physiological conditions is blunted by undernutrition. Fetal growth is dependent on the capacity of the mother and placenta to deliver nutrients. In our model, lack of nutrients was the same for male and female fetuses, and therefore, differences in placental functioning may contribute to the observed sex differences [29]. We found that in MUN females, placental weight was smaller (i.e., the placenta grew in proportion to fetal weight, whereas in males, placental weight enlarged proportionally more). Consequently, a lower feto-placental ratio, a marker of placental efficiency [30], was found in MUN males. Under nutrient shortage, the placenta can compensate malnutrition by surface expansion together with upregulation of transporters to maintain the nutrient supply. However, this strategy may be inadequate, since a growing fetus has to share nutrients with the enlarged placenta [15,31]. In our experimental animal model, the female MUN placenta grew less, a strategy that maintains efficiency, as evidenced by no alteration in feto-placental ratio. In humans, female placenta has also been found to develop more slowly in situations of low nutrient availability [32], and in rodents, it has been demonstrated to be more sensitive than male placenta to nutritional perturbations, with larger modification in gene expression [33]. In conclusion, our data highlight sex-specific differences in placental growth strategies under suboptimal nutrition, with worse adaptation in males (Figure 9).

Reduced placental efficiency may be the result of poor vascular development. We evaluated PlGF level and VEGF expression, which are important factors for placental growth and angiogenesis [34,35]. A marked decrease in placental VEGF expression was found in MUN males but not in females. These data suggest that male placenta may have deficient vascular growth. Poor vascularization of the placenta has been associated with pregnancy complications, such as maternal obesity, preeclampsia, and FGR. Reduced angiogenesis has also been detected in placenta from rats exposed to toxic substances [36] and in obesity models [37], showing reduced gene expression of VEGF and other angiogenic factors. However, these studies did not evaluate sex differences. We evaluated angiogenic factors at the end of gestation, and it is likely their levels should be declining as most of the placental vasculature has been completed. This is a limitation of the study, and histological assessment of the level of vascularization would enable us to conclude on sex differences under normal or stress conditions.

The lower VEGF expression in MUN male placenta might be related to excess ROS production since oxidative stress decreases the expression of several transcription factors mediating angiogenesis and has been implicated in pregnancy disorders [38]. Similar findings have been described in a pig model of maternal obesity during gestation, where decreased placental angiogenesis and expression of VEGF were associated with NADPH upregulation [37]. Conversely, we did not observe a significant difference in VEGF expression between MUN and control females and a lower XO expression, suggesting lower ROS production. Therefore, better placental vascularization, mediated by VEGF, could be one of the possible mechanisms responsible for the observed better placental efficiency and fetal growth in females under the same nutrient-restricted conditions.

Besides the inhibitory effect of ROS on angiogenic factors, the altered ROS–antioxidant balance observed in MUN male placenta can directly contribute to reduced placental efficiency, as occurs in pregnancy complications related to placental dysfunction [39]. The major ROS-producing and ROS-degrading antioxidant enzymes are expressed in the placenta [40]. MUN male placenta exhibited increased XO expression and lower levels of several antioxidant enzymes, namely, Mn-SOD, catalase, and UCP2. On the other hand, females responded to nutrient restriction with a better expression pattern, being Ec-SOD the only system downregulated, but catalase was increased and the ROS-producing enzyme XO was downregulated. Our data suggest that the oxidative disbalance observed in male placenta would lead to excess superoxide anion (Figure 9). This is a short-lived ROS. However, it will scavenge nitric oxide (NO), leading to the formation of peroxynitrite, a stable and highly oxidizing ROS [40]. This will create a vicious cycle, since peroxynitrite, through protein nitration, inhibits SOD activity, exacerbating oxidative stress [41]. Other potential targets for nitration with critical functions in the placenta can be nutrient transporters [42]. Mitochondria is a major site for peroxynitrite formation due to NO production by mtNOS and superoxide anion formation from mitochondrial electron transport chain leaks [42]. We suggest that the reduction in Mn-SOD and UCP2, relevant mitochondrial antioxidant enzymes [43] in MUN male placenta, would also lead to oxidative and nitrosative damage to the mitochondria. In addition, NO is one of the main vasodilators in placental vasculature, and a reduction would further compromise nutrient and oxygen delivery to the fetus [42]. Placental oxidative damage has been commonly found in rat models of fetal programming induced by several developmental insults, including maternal obesity [37], hypoxia [44,45], low protein diet [46], and in rats exposure to toxics during gestation [36]. Under our experimental conditions, oxidative damage to lipids tended to be higher in the placentas of both MUN males and females. We were surprised that females had the same level of lipid oxidative damage compared with males. It is possible that differences in nitrosative markers may exhibit sexual dimorphism, or specific mitochondrial alterations may also differ between sexes. In late gestation, exposure to hypoxia has been shown to decrease CI/CIV activity rates and increase nitration in male but not female placentas [47]. Whether these alterations also occur under maternal nutrient restriction deserves further attention.

GCs have important influences on maternal and fetal metabolism [48], and excess cortisol access to the fetus can reduce growth, as suggested by the inverse relationship between maternal cortisol levels and birth weight we reported in twin pregnancies [4]. Under physiological conditions, the fetus is protected from exposure to excess GCs by the placental enzymatic barrier 11β-HSD2. This enzyme is synthesized in the syncytial layer of the placental villi and converts active GCs into inactive forms (in rodents, it oxidizes and deactivates corticosterone to 11-dehydrocorticosterone). Therefore, fetal exposure to maternal GCs depends on the amount of this placental enzyme [14], and adverse conditions such as maternal stress, hypoxia, nutritional restriction [15,16,17], and toxic substances [36] can reduce its expression. In the present study, we observed an elevation of MUN maternal corticosterone, accompanied by a reduction of placental 11β-HSD2. Expression of the barrier was lower in male placenta, and accordingly, higher levels of corticosterone were detected in male fetuses. Females also exhibited a lower enzyme expression compared with controls, but female fetuses did not show a significant elevation of fetal plasma corticosterone, suggesting that the level of 11β-HSD2 expression was sufficient (Figure 9). The precise mechanism of maternal malnutrition downregulation of placental 11β-HSD2 expression remains unclear and deserves further attention. Among others, hypermethylation on the 11β-HSD2 promoter has been put forward [14,49], and it has been demonstrated in humans with FGR [49].

It has been demonstrated that excess maternal corticosterone increases placental oxidative stress, impairs vascularization, and reduces the expression of nutrient transporters [10,12,13,20]. Moreover, exposure to increased maternal GCs impairs normal placental vasculature development by decreasing VEGF mRNA expression [13]. Based on these findings, we propose that increased maternal corticosterone, as a result of the stress imposed by undernutrition, may be the initial stimulus responsible for the observed placental alterations in oxidative balance and VEGF expression. The differences in how male and female placentas respond to the excess GCs may be responsible for the sexual dimorphism in placental efficiency.

## 4. Materials and Methods

### 4.1. Maternal Undernutrition (MUN) Model

Experiments were performed on Sprague Dawley rats from the Animal House at the Universidad Autónoma de Madrid (ES-28079-0000097). All experimental procedures conformed to the Declaration of Helsinki, the Guidelines for the Care and Use of Laboratory Animals (NIH publication No. 85-23, revised in 1996), and the Spanish legislation (RD 1201/2005). The experiments were approved by the Ethics Review Board of the Universidad Autónoma de Madrid (CEI-UAM 96-1776-A286) and the Regional Environment Committee of the Comunidad Autónoma de Madrid (RD 53/2013; Ref. PROEX 04/19).

The welfare of the animals was monitored by specialized staff at the Animal House. The veterinary in charge certified that the rats were free from pathogens that might interfere with the experiments. The rats were housed in 36.5 × 21.5 × 18.5 cm (length/width/height) buckets on aspen wood and kept with a 12 h/12 h light–dark photoperiod at a constant temperature of 22 °C and 40% relative humidity.

A model of fetal programming induced by maternal undernutrition during gestation (MUN) was used, as previously described [50]. The day sperm was observed in the vaginal smear was considered the first day of gestation (G0), and the pregnant rats were allocated to one of the two experimental groups (control or MUN). Control rats were fed *ad libitum* throughout gestation. MUN rats were fed *ad libitum* from G0 to G10, and from G11 to G20, food was restricted to 50% of the averaged control daily intake. Breeding diet was as follows: 55% carbohydrates, 22% protein, 4.4% fat, and 4.1% fiber (Euro Rodent Breeding Diet 22, 5LF5, Labdiet, Madrid, Spain). Drinking water was provided *ad libitum* throughout the study.

### 4.2. Experimental Design

All the experiments were performed at G20; eight MUN and eight control dams and their litter were used. The dam was weighted on a scale (Denver Instrument SI-8801, Denver Instrument, Arvada, CO, USA). Thereafter, it was anaesthetized in a CO_2_ chamber. The blood sample was obtained by cardiac puncture in heparin-coated tubes (ROVI, Madrid, Spain; 5.000 U/mL). A digital glucometer (Accu-Check Aviva, Roche, Barcelona, Spain) was used to assess maternal plasma glucose concentration (mg/dL). All extractions were performed at 2:00 p.m. Thereafter, the dam was killed by excess CO_2_, and a laparotomy was performed to dissect the uterine horns and to obtain individual fetus and placenta pairs.

The fetuses were sexed by evaluating the distance between the genital papilla and anal aperture, which is larger in males than in females. From each litter, three to four male and three to four female feto-placenta pairs were randomly chosen. The fetuses were weighted with a precision scale (PCE-AB Class l, PCE Instruments, Alicante, Spain), and their lengths were measured with a digital caliper (Conecta, Nessler, Madrid, Spain).

The fetuses were killed by decapitation, and trunk blood was collected in heparin tubes. The blood from two to three fetuses of the same sex and litter was pooled.

Immediately after extraction, maternal or fetal blood was centrifuged at 900× *g* for 10 min at 4 °C to obtain plasma, which was aliquoted and stored at −80 °C until use.

### 4.3. Oxidative Status in Plasma and Placenta

Placental tissue was homogenized with 20 mM Tris buffer (pH 7.4). Thereafter, the tissue was centrifuged at 10,000 rpm at 4 °C for 10 min, and the supernatant aliquoted and stored at −80 °C. Bradford assay (Bio-Rad, Pleasanton, CA, USA) was used to assess maternal plasma proteins, and absorbance was measured in a plate reader (Synergy HT Multi-Mode Microplate Reader, BioTek, Winooski, VT, USA).

Oxidative damage to lipids was assessed by a Lipid Hydroperoxide Assay Kit (LPO, Bioquochem, Gijón, Spain), which analyzes malondialdehyde (MDA) + hydroxynonenal (HNE) concentrations, as previously described [47]. The experiments were performed according to the manufacturer’s instructions. Absorbance was measured in a microplate reader. MDA + HNE content was expressed as Μm.

Protein carbonyls. Oxidative damage to proteins was assessed using a 2,4-dinitrophenylhydrazine assay, which detects protein carbonyls, adapted to a microplate reader, using the extinction coefficient of 2,4-dinitrophenylhydrazine (ε 22,000 M/cm), as previously described [46]. Absorbance was measured at 595 nm in a microplate reader, and the data were expressed as nmol/mg protein.

Reduced glutathione (GSH). Plasma GSH was assessed by a fluorimetric micromethod based on the reaction with ophthalaldehyde, as previously described [46]. Fluorescence was measured in a Synergy HT Multi-Mode Microplate Reader at 360 ± 40 nm excitation and 460 ± 40 nm emission wavelengths. The GSH level of the samples was expressed as mg/mg protein.

Total thiols. Plasma thiol groups were assessed by a 5,5’-dithiobis (2-nitrobenzoic acid) assay, adapted to a microplate reader, as previously described [51]. Absorbance was measured at 412 nm, and thiol content was expressed as mM/mg protein.

Superoxide dismutase (SOD) activity. SOD activity was assessed by a kit (SOD Activity Assay kit, KB-03-011, Bioquochem, Gijón, Spain) as previously described [47]. The experiments were performed according to the manufacturer’s instructions. Absorbance was read at 450 nm in a plate reader. SOD activity was expressed as mg/mg protein.

Catalase activity. Catalase activity was assessed by a kit (KB-03012, Bioquochem, Gijón, Spain). Absorbance was measured at 540 nm in a microplate reader. Catalase activity was expressed as U/mg protein.

### 4.4. Western Blot Assay in Placental Tissue

Placental tissue was homogenized with lysis buffer of the following composition: 0.42 mM NaCl, 1 mM Na_4_P_2_O_7_, 1 mM DTT, 20 mM HEPES, 20 mM NaF, 1 mM Na_3_VO_4_, 1 mM EDTA, 1 mM EGTA, 20% glycerol, 2 mM phenylmethylsulfonyl fluoride (PMSF), 1 μL/mL leupeptin, 1 μL/mL aprotinin, and 0.5 μL/mL N-alpha-p-Tosyl-L-lysine chloromethyl ketone hydrochloride (TLCK- hydrochloride). Thereafter, the tissue was centrifuged for 30 min at 10,000 rpm and 4 °C, and the supernatant was collected. Bradford assay was used to assess the protein level in the samples, which were stored at −80 °C for later analysis.

Western blotting was performed as previously described [50]. Briefly, 30 µg protein samples were separated by 15% SDS-PAGE gels. Primary antibodies against p22^phox^ (Santa Cruz Biotechnology, Heidelberg, Germany; 1:400 final dilution), Cu/Zn-SOD, and Mn-SOD (Santa Cruz Biotechnology, Heidelberg, Germany; 1:1000 final dilution); EC-SOD (Enzo Life Sciences, Farmingdale, NY, USA; 1:1000 final dilution); XO (Santa Cruz Biotechnology, Heidelberg, Germany; 1:1000 final dilution) and catalase (Sigma-Aldrich, St. Louis, MO, USA; 1:2000 final dilution); UCP2 (Santa Cruz Biotechnology, Heidelberg, Germany; 1:1000 final dilution); VEGF (Santa Cruz Biotechnology, Heidelberg, Germany; 1:500 final dilution); and 11β-HSD2 (Sigma-Aldrich, St. Louis, MO, USA; 1:500 final dilution) were applied overnight at 4 °C.

After washing, secondary antibodies (antirabbit or antimouse IgG peroxidase conjugated) were applied for 1 h. Blots were washed and incubated in commercially enhanced chemiluminescence reagents (ECL Prime, Amersham Bioscience, Little Chalfont, Buckinghamshire, UK), and bands were detected by ChemiDoc XRS+ imaging system (Bio-Rad, USA).

Blots were re-incubated with β-actin antibody (Santa Cruz Biotechnology, Heidelberg, Germany; 1:4000 final dilution). They were quantified using Image Lab 3.0 software (Bio-Rad, USA). Expression values of p22^phox^, Cu/Zn-SOD, Mn-SOD, EC-SOD, XO, catalase, UCP2, VEGF, and 11β-HSD2 were normalized with β-actin to account for variations in gel loading.

### 4.5. Corticosterone Assay in Plasma

Plasma corticosterone was assessed by a competitive enzyme immunoassay kit according to the manufacturer’s instructions (Corticosterone Rat/Mouse ELISA kit, DV9922, Demeditec, Kiel, Germany).

### 4.6. Placental Growth Factor (PlGF)

PlGF was analyzed by ELISA (Rat Placenta Growth Factor ELISA kit, Abbexa, Cambridge, UK) according to the manufacturer’s instructions. Absorbance was detected at 450 nm in a plate reader and expressed as pg/mL.

### 4.7. Statistical Analysis

Statistical analysis was performed with GraphPad Prism (version 5, San Diego, CA, USA). The Kolmogorov–Smirnov test was used to analyze the normality of the data. Data with normal distribution were expressed as mean ± standard error of mean (SEM), and they were analyzed by Student´s *t*-test or by one-way ANOVA with Bonferroni correction for multiple comparisons. Those data that did not follow a normal distribution were expressed as median and interquartile range, and they were analyzed by the Kruskal–Wallis test followed by Dunn’s test. Significant differences were established at *p*-value < 0.05.

## 5. Conclusions

Suboptimal nutrition during gestation induces stress response and elevates maternal corticosterone.Placental adaptation to undernutrition exhibits sexual dimorphism, showing reduced efficiency in males.Lower efficiency in MUN male placenta may be related to excess ROS and reduced angiogenic factors and GC barrier, and may contribute to higher susceptibility to cardiovascular disease programming.Female placenta has better adaptive response to maternal undernutrition with moderate growth and better oxidative balance, angiogenesis, and GC barrierLower GC access to female fetus can contribute to lower susceptibility to cardiovascular disease programming in response to undernutrition.

## Figures and Tables

**Figure 1 ijms-22-00237-f001:**
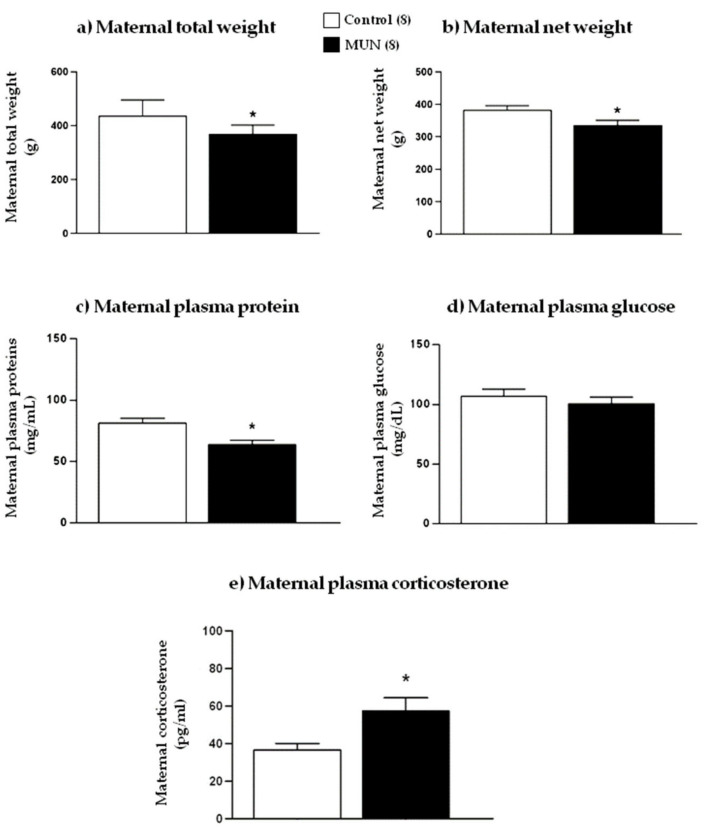
(**a**) Maternal total weight, (**b**) net weight, (**c**) plasma protein, (**d**) plasma glucose, and (**e**) plasma corticosterone of dams fed *ad libitum* (control) and dams exposed to undernutrition during pregnancy (MUN). Data are presented as mean ± SEM; sample size is shown in the legend; * *p*-value < 0.05.

**Figure 2 ijms-22-00237-f002:**
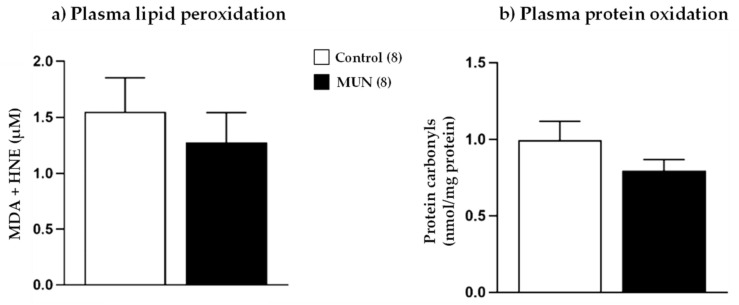
Maternal plasma levels of (**a**) lipid peroxidation and (**b**) protein oxidation of dams fed *ad libitum* (control) and dams exposed to undernutrition during pregnancy (MUN). Data are shown as mean ± SEM; sample size is shown in the legend.

**Figure 3 ijms-22-00237-f003:**
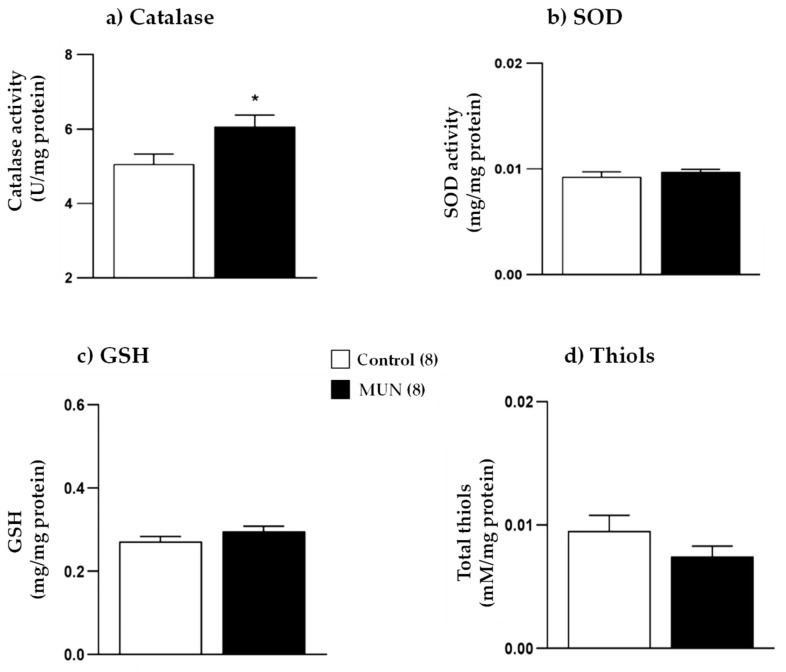
Maternal plasma (**a**) catalase activity, (**b**) SOD activity, (**c**) GSH, and (**d**) thiol groups from dams fed *ad libitum* (control) and dams exposed to undernutrition during pregnancy (MUN). Data are shown as mean ± SEM; sample size is shown in the legend; * *p*-value < 0.05.

**Figure 4 ijms-22-00237-f004:**
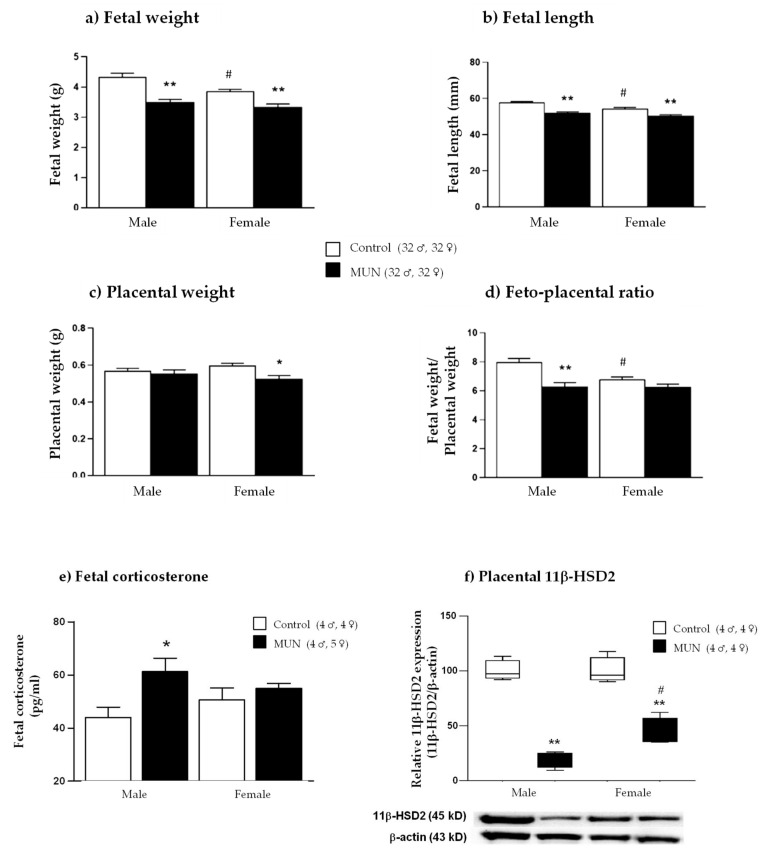
(**a**) Fetal weight, (**b**) fetal length, (**c**) placental weight, (**d**) feto-placental ratio, (**e**) fetal plasma corticosterone, and (**f**) placental expression of 11β-HSD2 in males and females from control and maternal undernutrition (MUN) groups. Data are shown as mean ± SEM except 11β-HSD2 expression, which is shown as median ± interquartile range; sample size is shown in the legend; * *p*-value < 0.05, ** *p*-value < 0.01 vs. sex-matched controls, # *p*-value < 0.05 vs. MUN males.

**Figure 5 ijms-22-00237-f005:**
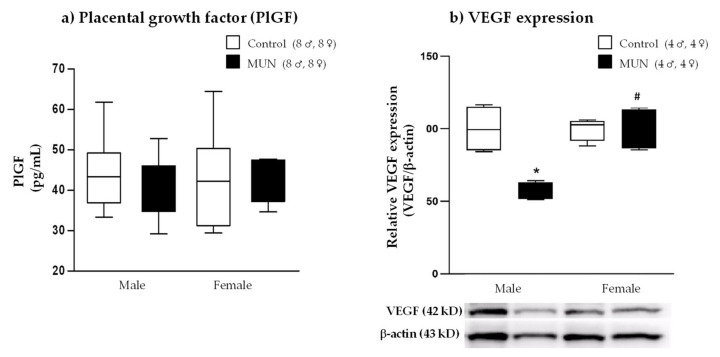
(**a**) Placental growth factor (PlGF) and (**b**) expression of vascular endothelial growth factor (VEGF) in placentas from control and maternal undernutrition (MUN) groups. Data are shown as median ± interquartile range; sample size is shown in the legend; * *p* < 0.05 vs. sex-matched control, # *p* < 0.05 vs. MUN males.

**Figure 6 ijms-22-00237-f006:**
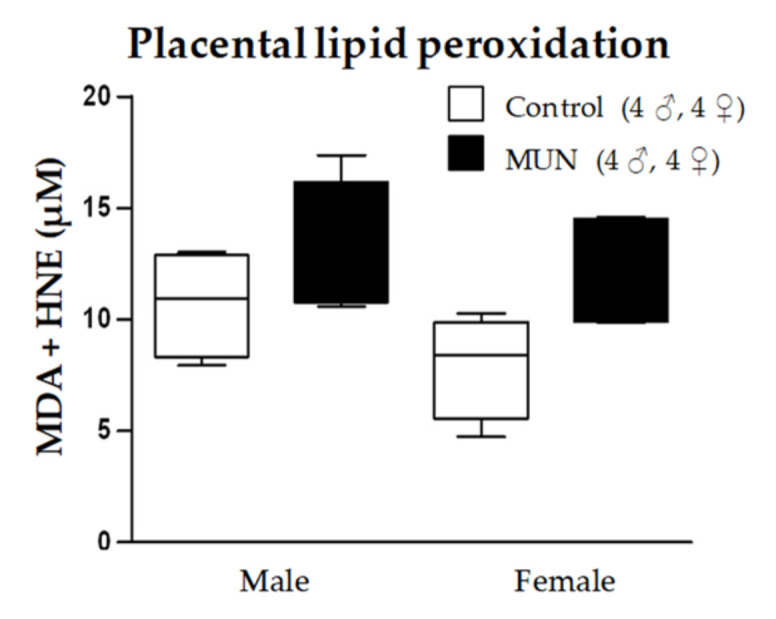
Lipid peroxidation in placentas from control and maternal undernutrition (MUN) groups. Data are shown as median ± interquartile range; sample size is shown in the legend.

**Figure 7 ijms-22-00237-f007:**
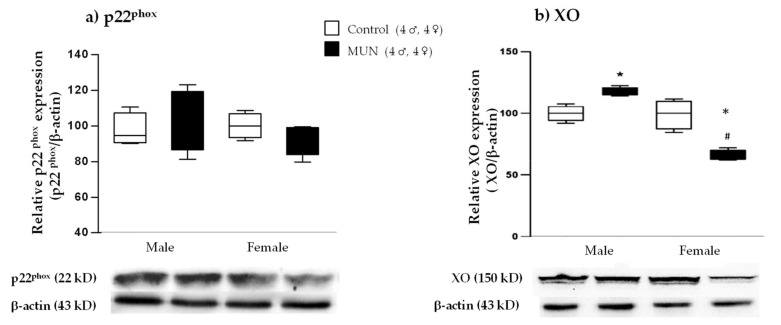
Placental relative expression of (**a**) NADPH oxidase subunit p22^phox^ and (**b**) xanthine oxidase (XO) from control (C) and maternal undernutrition (MUN) groups. Data are shown as median ± interquartile; sample size of dams is shown in the legend; * *p*-value < 0.05 vs. sex-matched control; # *p*-value < 0.05 vs. MUN males.

**Figure 8 ijms-22-00237-f008:**
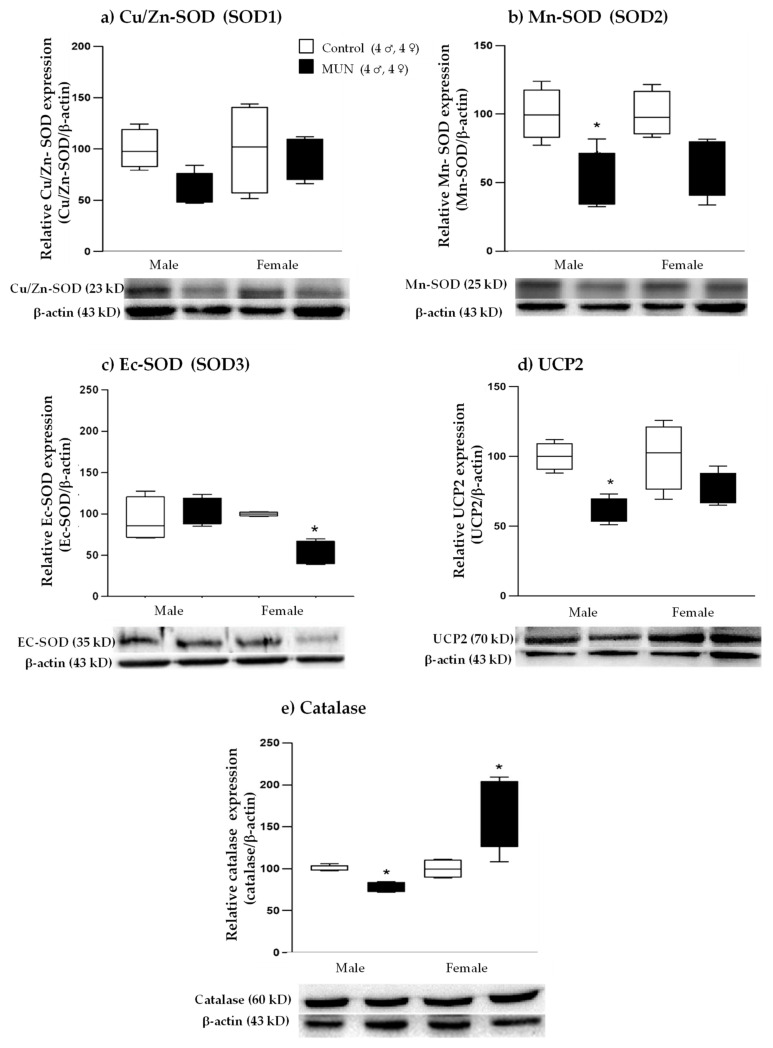
Placental relative expression of (**a**) Cu/Zn-SOD (SOD1), (**b**) Mn-SOD (SOD2), (**c**) Ec-SOD (SOD3), (**d**) uncouple protein 2 (UCP2), and (**e**) catalase from control and maternal undernutrition (MUN) groups. Data are shown as median ± interquartile range; sample size of dams is shown in the legend; * *p*-value < 0.05 vs. sex-matched controls.

**Figure 9 ijms-22-00237-f009:**
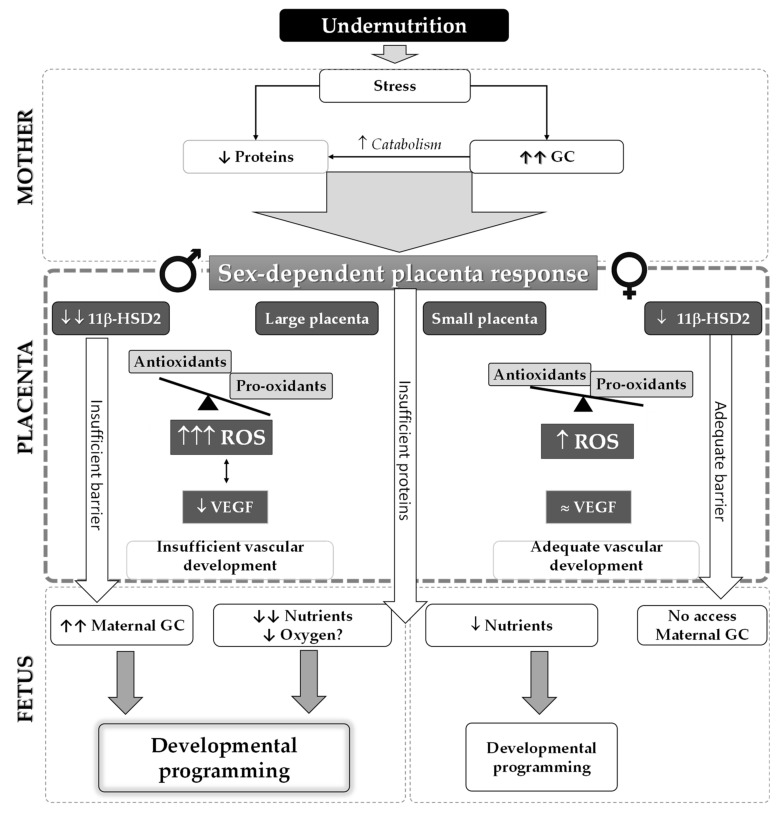
Schematic representation of the main findings and possible mechanisms explaining sex differences in placental adaptation to maternal undernutrition. Insufficient nutrition leads to maternal stress and release of corticosterone, increasing catabolism and ROS. Placental response exhibits a sexual dimorphic pattern. Male placenta does not reduce growth, competing with the fetus for nutrients. It upregulates ROS-producing enzymes and downregulates ROS-degrading enzymes (including mitochondrial antioxidants), leading to high ROS milieu. Excess ROS may contribute to the observed downregulation of VEGF protein expression, reducing angiogenesis, and possibly inadequate vascular development, compromising nutrient and oxygen access to the fetus. Downregulation of 11β-HSD2 increases GC access to the fetus, further contributing to programming. Female placenta reacts to low nutrients and a high GC environment with moderate placental growth, more balanced oxidative pattern, higher expression of 11β-HSD2, and normal VEGF. These adaptations would ensure better access of nutrients, avoiding the passage of GCs to the fetus. GCs, glucocorticoids; ROS, reactive oxygen species; VEGF, vascular endothelial growth factor; 11β-HSD2, 11β-hydroxysteroid dehydrogenase type 2; arrows represent elevations or reductions, the scale represents oxidative balance.

## Data Availability

Raw data can be provided to researchers interested on request to the corresponding author.

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
