# Peer review of "Sex Differences in Placental Protein Expression and Efficiency in a Rat Model of Fetal Programming Induced by Maternal Undernutrition"

_ijms, 2020, doi:10.3390/ijms22010237_

Round 1

Reviewer 1 Report

Phuthong et al., present a well written paper outlining the effects of undernutrition on Maternal, Placental and Fetal health. This work is well founded and adds to our understanding on how fetal outcome sex differences may arise under conditions of maternal stress.

Specific comments:

Abstract: It is concise but lacks context at the start that would set the need for the work.

Results: I would suggest significant editing by the authors. Numerical data are not described in the narrative with significance level indicated making it difficult to read and understand the magnitude of the changes in treatment groups. Often the purpose/question the data is answering is mixed in with the data description. It may be easier to follow if it was presented first. The few discussional statements included should be reserved for the discussion.

Figure 9 describes maternal and fetal corticosterone production after the presentation of all of the placenta data. Consider moving this to be in sequence with the rest of the maternal and fetal data.

Figure 10: It somewhat overwhelming but does fairly well at representing the complexities of the data. Attempts to simplify should be considered. It should be referred to throughout the discussion not in the first and last paragraphs alone.

The detailed statistical analysis descriptions in the figure legends should be removed as this is described in the methods.

Conclusions: There is no conclusion that eludes to the findings in the female fetus.

Author Response

Phuthong et al., present a well written paper outlining the effects of undernutrition on Maternal, Placental and Fetal health. This work is well founded and adds to our understanding on how fetal outcome sex differences may arise under conditions of maternal stress.

Specific comments:

  • Abstract: It is concise but lacks context at the start that would set the need for the work.

ANSWER: We have added a sentence at the beginning of the abstract to set the context of the study (lines 20-22).

  • Results: I would suggest significant editing by the authors. Numerical data are not described in the narrative with significance level indicated making it difficult to read and understand the magnitude of the changes in treatment groups. Often the purpose/question the data is answering is mixed in with the data description. It may be easier to follow if it was presented first. The few discussional statements included should be reserved for the discussion.

ANSWER: We have included the statistical significance level in the text and excluded the discussion statements according to reviewer suggestions.

  • Figure 9 describes maternal and fetal corticosterone production after the presentation of all of the placenta data. Consider moving this to be in sequence with the rest of the maternal and fetal data.

ANSWER: We have included maternal corticosterone in figure 1e, and also moved fetal corticosterone and 11-βHSD2 expression to figure 4e-4f. Therefore, there is one figure less in the paper.

  • Figure 10: It somewhat overwhelming but does fairly well at representing the complexities of the data. Attempts to simplify should be considered. It should be referred to throughout the discussion not in the first and last paragraphs alone.

ANSWER: We have tried to simplify figure 10 (now figure 9) and referred to it throughout the discussion, when we considered it was useful.

  • The detailed statistical analysis descriptions in the figure legends should be removed as this is described in the methods.

ANSWER: We have excluded the details of statistical analysis in all the figures

  • Conclusions: There is no conclusion that eludes to the findings in the female fetus.

ANSWER: We have now modified the conclusions, including also those related to females.

Reviewer 2 Report

Please find my comments below:

  1. Why do the authors refer to the increase in maternal catalase as a compensatory response? If so, did they see a similar increase in other anti-oxidant enzymes? If not, why just catalase? These points can be addressed in the discussion.
  2. Figure 4: The differences observed seem to be very minimal, can the authors present the data with SD?
  3. What is the significance of lower expression of SOD2 but not SOD3?
  4. The placental PlGF and VEGF levels were assessed at G20 which is almost the end of gestation and levels of these growth factors should be declining as most of the placental vasculature & growth has already been completed.
    A better way to compare placental vascularization/growth would be histological studies of placental sections.

Author Response

Please find my comments below:

  1. Why do the authors refer to the increase in maternal catalase as a compensatory response? If so, did they see a similar increase in other anti-oxidant enzymes? If not, why just catalase? These points can be addressed in the discussion.

ANSWER: We only found an increase in catalase activity, but not in other antioxidants. We have discussed a possible explanation for this finding and compared it to recent meta-analysis analyzing the different antioxidants in serum from women with preclampsia [1]. This study evidences that catalase may be an enzyme which increases in response to oxidative stress. Instead, SOD is less responsive and may be even inactivated by H2O2 [2]. We have now addressed this aspect in the discussion (line 231-241).

  1. Figure 4: The differences observed seem to be very minimal, can the authors present the data with SD?

ANSWER: We think the statistical differences in fetal anthropometric parameters and placenta are robust, since they are in the range of p-values=0.001 and 0.0001 (now included in text). SD express the variability of data, while SEM shows the dispersion of a data relative to its sample size. When the variable is being analyzed by a parametric-test, the SEM could be more suitable, because the significant probability depends on the sample size between groups. In Figure 4 we have a large sample size and we think the mean estimates the true mean of the population with a good precision. Therefore, we suggest maintaining SEM in the figure. Besides, it would be confusing to describe some figures with SEM and others with SD.

Below we have included a table with SD in order to express the variability of data. We have also included the 95% confidence interval, which could show the mean accuracy in the range. If you consider this table is relevant, we can include it as supplemental material.

Table 1. Fetal anthropometric parameters and placenta from MUN and control rats

Male control

(n=32)

Male MUN

(n=32)

Female control

(n=32)

Female MUN

(n=32)

Fetal weight (g)

4.33±0.68

[4.088-4.581]

3.49±0.52

[3.303-3.683]

3.85±0.38

[3.714-3.991]

3.33±0.58

[3.127-3.539]

Fetal length (mm)

57.71±3.93

[56.38-59.04]

52.10±2.47

[51.08-53.12]

54.46±3.15

[53.18-55.73]

50.54±2.67

[49.64-51.45]

Placental weight (g)

0.57±0.07

[0.544-0.594]

0.55±0.12

[0.513-0,598]

0.59±0.07

[0.571-0.622]

0.52±0.10

[0.487-0.562]

Fetal weight/

placenta weight

7.98±1.46

[7.451-8.507]

6.27±1.62

[5.686-6.853]

6.77±1.03

[6.399-7.141]

6.22±1.15

[5.796-6.644]

Data are shown as mean ± SD; confidence intervals are shown in brackets. C, control; MUN, maternal undernutrition.

  1. What is the significance of lower expression of SOD2 but not SOD3?

ANSWER: The primary location of SOD3 is the extracellular matrix and on cell surfaces and it regulates extracellular levels of superoxide anion while SOD2 is located in the mitochondria [2]. Mitochondria are major sites of ROS production. Therefore, if SOD2 expression is low, mitochondria are also a target for oxidative/nitrosative damage, producing further dysregulation (ROS-induced ROS release). Therefore, we suggest that a lower expression of SOD2 may damage mitochondria and be particularly detrimental for placental function. We have highlighted the importance of mitochondria damage in the text (lines 299-301)

  1. The placental PlGF and VEGF levels were assessed at G20 which is almost the end of gestation and levels of these growth factors should be declining as most of the placental vasculature & growth has already been completed. A better way to compare placental vascularization/growth would be histological studies of placental sections

ANSWER: We agree on the point raised by the reviewer and how was the vasculature developed in MUN and control males and females deserves further analysis. We have now included this aspect as a limitation of the study in discussion section (line 272-275).

References:

  1. Taravati A, Taiwan TF: Comprehensive analysis of oxidative stress markers and antioxidants status in preeclampsia. . J Obstet Gynecol. 2018, 57(6):779-790.
  2. Fukai T, M. U-F: Superoxide dismutases: role in redox signaling, vascular function, and diseases. Antioxid Redox Signal. 2011, 15(6):1583-1606.